# Highways to happiness for autistic adults? Perceived causal relations among clinicians

**Marie K. Deserno**[1,2]*, **Denny Borsboom**[2], **Sander Begeer**[3], **Riet van Bork**[2], **Max Hinne**[2], **Hilde M. Geurts**[1,2]

**1** Dr. Leo Kannerhuis and REACH-AUT, Doorwerth, The Netherlands, **2** Department of Psychology, University of Amsterdam, Amsterdam, The Netherlands, **3** Section Clinical Developmental Psychology, Vrije Universiteit Amsterdam, Amsterdam, The Netherlands

* marie.deserno@gmail.com

**Data Availability Statement:** All relevant data are within the paper and its Supporting information files.

## Abstract

The *network approach* to psychological phenomena advances our understanding of the interrelations between autism and well-being. We use the Perceived Causal Relations methodology in order to (i) identify perceived causal pathways in the well-being system, (ii) validate networks based on self-report data, and (iii) quantify and integrate clinical expertise in autism research. Trained clinicians served as raters (N = 29) completing 374 cause-effects ratings of 34 variables on well-being and symptomatology. A subgroup (N = 16) of raters chose intervention targets in the resulting network which we found to match the respective centrality of nodes. Clinicians' perception of causal relations was similar to the interrelatedness found in self-reported client data (N = 323). We present a useful tool for translating clinical expertise into quantitative information enabling future research to integrate this in scientific studies.

## Introduction

An Autism Spectrum Disorder diagnosis (ASD) is related to reduced levels of well-being [1–3], but the study of risk and protective factors for well-being in autistic individuals is still at an early stage, in particular for adults. After at least a decade of well-being research in autistic adults, we lack sophisticated understanding of the potential causal pathways that channel the heterogeneity in adult outcome of the autistic population. Most research on later outcome and well-being in autistic individuals aspires to the principle of Ockham's razor, looking for a set of simple basic elements or a latent entity that could explain the emerging phenomenological complexity. In addition, most of these attempts study the interrelatedness of two or three variables instead of the complex system of possible pathways that extend further than just two variables.

### Network approach to psychological phenomena

Recent theoretical literature, however, has argued for a more complex approach to human development inspired by dynamical systems theory [4–7]. This theoretical shift has enabled

**Funding:** DB is supported by ERC Consolidator Grant no. 647209. HMG is supported by the NWO VICI Grant no. 453-16-006. This research project is supported by ZonMW Grant no. 70-73400-98-002.

**Competing interests:** The authors have declared that no competing interests exist.

the emergence of the *network approach*, an alternative psychometric conceptualization in which psychological phenomena are seen as a dynamic set of causally intertwined properties [8, 9]. The starting point in the network approach is to determine the relational structure between symptoms and other factors and to represent this information in a network. Currently, there are at least four approaches to the construction of these networks. First, by examining associations between symptoms in a population [10]; second, by inspecting the dynamic structure of a network over time [11]; third, by utilizing the structure of diagnostic manuals [12, 13]; and fourth, by eliciting judgements on the structure of causal relations between symptoms, either from clinicians [9, 14] or through self-report [15]. In our recent network studies, we have studied the association network of interacting factors for the subjective well-being of autistic individuals [16, 17] based on self-report survey data. However, the inclusion of clinical expertise is largely lacking. Furthermore, due to the explorative character of network analysis as a statistical method, it is unclear whether relations between variables identified in these studies actually reflect causal interactions (rather than, e.g., the effect of unmeasured common causes). In addition, it remains unclear what the direction of these causal pathways is, and to what extent the relevant pathways are also identifiable by other, independent modes of observation. The primary purpose of this study is to quantify multivariate causal beliefs of clinicians specialized in autism care and to map those onto multivariate empirical interrelations found in self-reported data of autistic adults, which will add to the understanding of causal relations between autistic characteristics and well-being.

## Perceived causal pathways in the well-being system

The general interest in *causal pathways* in the well-being system is a crucial issue. Most of the network literature, however, focuses on the first of the above-mentioned methodologies: the analysis of association structures in population data. Although association networks are a good way to develop insight into the association structures among symptoms, to be able to develop interventions and policy based on network information on ASD and well-being, we need to gain more insight into the directionality of the interrelations. For example, if cognitive problems associated with autism are connected to depressed mood in an association network, does that mean that specific cognitive problems associated with ASD cause depressed mood? Alternatively, does depressed mood cause cognitive problems associated with ASD, or are the variables connected in a feedback loop? That is, the field is in dire need for a toolbox that can help us determine which connections in the network represent directed causal effects that arise from reciprocal causation or coupled equilibria, and which associations are due to the effect of unmeasured variables.

## Clinical validation of networks based on self-report data

From earlier research we have learned that, when asked how symptoms relate to each other, clinicians report a network of interacting symptoms [14, 18]. This suggests that professionals working in the clinical field may already conceptualize psychological constructs as a set of causal relations between symptoms and other factors. A brief review of the existing network literature shows that this source of information has not yet been integrated in network approaches. It remains unclear, for instance, whether the network structures shown in self-reported data actually resemble those networks that clinicians would report. In other words, when compared to the networks based on self-reported data from earlier studies on factors relevant for well-being in ASD (e.g., insistence on sameness, experiencing reduced contact and struggling with social conventions; [17]), do experienced clinicians report a similar pattern of causal relations between these factors? And beyond that: do clinical professionals, based on

their own experience and knowledge, intuitively choose to intervene on those factors that a network analysis would reveal as most influential in the network?

## Integration of clinical expertise into empirical network studies

Even though many researchers aim to bridge the gap between scientific studies and clinical experience, they struggle to find a way to integrate the knowledge of experienced clinicians. One prominent reason for this might be a lack of tools to examine clinician experience in a structured way. Usually, this type of investigation would result in qualitative data from face-to-face interviews with clinicians, which is extremely important but difficult to integrate with common analytical tools in quantitative psychological science, which are statistical in nature. Recent studies, for example, have used the Delphi methodology (a structured interview/communication technique) to investigate the array of clinical practices used in the ASD realm [19, 20]. When exploring the causal relations in a network of symptoms and other clinically relevant factors, however, urgent questions remain: How can we represent the qualitative information in a formal system so that we can integrate the knowledge of experienced clinicians into the network framework? Can methods that elicit such expert knowledge be combined with network analysis of survey data to obtain a better picture of the structure of a problem domain?

## The current study

The first aim of the current study is to address the questions raised above, by constructing a symptom network on the basis of expert judgments [15] to visualize the relationships among characteristics of ASD and multiple facets of outcome and well-being. The second aim is to combine this information from clinical experts with the empirical networks obtained from statistical analyses of survey data. To this end, we utilize the methodology of Perceived Causal Relations scaling (PCR; [15]), which provides simple yet promising tools to assess perceived causal relations between variables, and combine this methodology with network analyses on self-reported data. In PCR scaling, any type of informant (i.e., rater) can be asked to what extent they attribute a causal relationship to a combination of a specific factor X and specific factor Y. Recent studies implementing this scaling technique have used it to get a self-reported representation of symptom-to-symptom interactions administered to individuals experiencing symptoms related to posttraumatic stress and anxiety [15], repetitive behaviors [14] and posttraumatic stress and eating disorders [21]. With this methodology, not only patients themselves but also knowledgeable experts can serve as raters who provide attributions concerning causal interrelationships between factors of interest [14].

In this exploratory study, we (i) identify clinicians' perceived causal relations between ASD characteristics and domains of well-being (as presented in [17]) and intervention targets within this causal network, (ii) investigate the resemblance of the clinician's perception of how factors in the network of ASD and well-being are interrelated and the association network based on the interrelations of these factors found in self-reported data, and (iii) provide an example of how to integrate knowledge of clinicians in empirical studies.

## Methods

### Participants

Experienced clinicians working with autistic adults will serve as raters of the constructed PCR scale which will enable us to calculate the inter-rater reliability for the causal belief network. Twenty-nine clinicians were included in the current study. These clinicians were selected

based on their years of experience with ASD to serve as *raters* in our rating task, i.e., the PCR scale. Potential raters were contacted through Dutch institutional networks such as the dr. Leo Kannerhuis (a nationwide specialized autism clinic), CASS18+ (the national network for autistic adult healthcare professionals), and institutions that are associated with the Academic Centre Reach-Aut, a collaborative network of autistic individuals, relatives, clinicians, and researchers. Since we distributed the questionnaire through the three leading clinical networks within the autism realm in the Netherlands and targeted specific individuals of which we knew that they are (clinically speaking) the authority in the Netherlands, we know that all clinicians participating in the current study are involved in diagnostic work, consultation, and intervention services. Also, these institutes all work with multidisciplinary teams as is recommended in the Dutch Multidisciplinary guidelines for ASD [22]. We included clinicians who are (1) an officially registered psychologist or psychiatrist, a behavioral scientist or social worker with (2) at least five years of clinical experience with adults in ASD health care. If a clinician decided to participate, they were asked to fill in the online informed consent following a link that we provided in an email. Raters were invited for the study online, and completed the PCR-assessment on their own via the internet at a place of convenience. Eventually, twenty-nine clinicians were recruited from various mental health institutions and universities to serve as raters in the current study. Most of them were clinical psychologists (N = 19), a smaller group indicated to work as a psychiatrist (N = 6) or another profession (N = 4). The majority of the participants were female (N = 21) with an average age of 48.5 (SD = 11, range: 30 to 66 years). The average clinical experience was twenty-three years (SD = 10, range: 7 to 50 years) of clinical work with, on average, 14 years (SD = 10, range: 4 to 50 years) of clinical experience in the autism realm. On average, they reported 22.5 hours of clinical work per week with hours ranging from 3 to 37 per week. The research reported in this manuscript has been approved by the Ethics Committee of the University of Amsterdam (2016-BC-7452).

## Measures and procedure

First, following the technique of Frewen et al. [15], perceived causal associations between a set of factors (presented below) were rated by experienced clinicians using a PCR scale. The scale requires a rating for each *direct* relation for each pair of items, which enables us to deduce perceived causal relationships between the network factors. The relations inferred from the PCR network were analysed and interpreted. Second, the PCR network was mapped and compared to the association network of the exact same variables based on self-report found in Deserno et al. [17]. In that study, we estimated network structures relating autism symptomatology to daily functioning and subjective well-being in 323 adult individuals with clinically identified autism (aged 17 to 70 years).

For the current study, we constructed a PCR scale with the online survey software Qualtrics (Qualtrics, Provo, UT) based on the 34-node-network based on self-reported data from an earlier study ([17]; see Fig 1) with a technique drawn from Frewen et al. [15]. In our earlier study, the resulting network was based on items (and subscales) from three measures relevant for well-being in ASD: the Adult Social Behavior Questionnaire (ASBQ; [23]), the Manchester Short Assessment of Quality of Life (MANSA; [24]), and the Health of the Nation Outcome Scales (HoNOS; [25]).

The ASBQ has been developed to yield an individual's self-reported score profile among six ASD domains: reduced empathy ($N_{items}$ = 7), reduced contact ($N_{items}$ = 7), reduced interpersonal insight ($N_{items}$ = 8), violation of social conventions ($N_{items}$ = 6) and insistence on sameness ($N_{items}$ = 8). We included all six subscales of the ASBQ in the PCR scale.

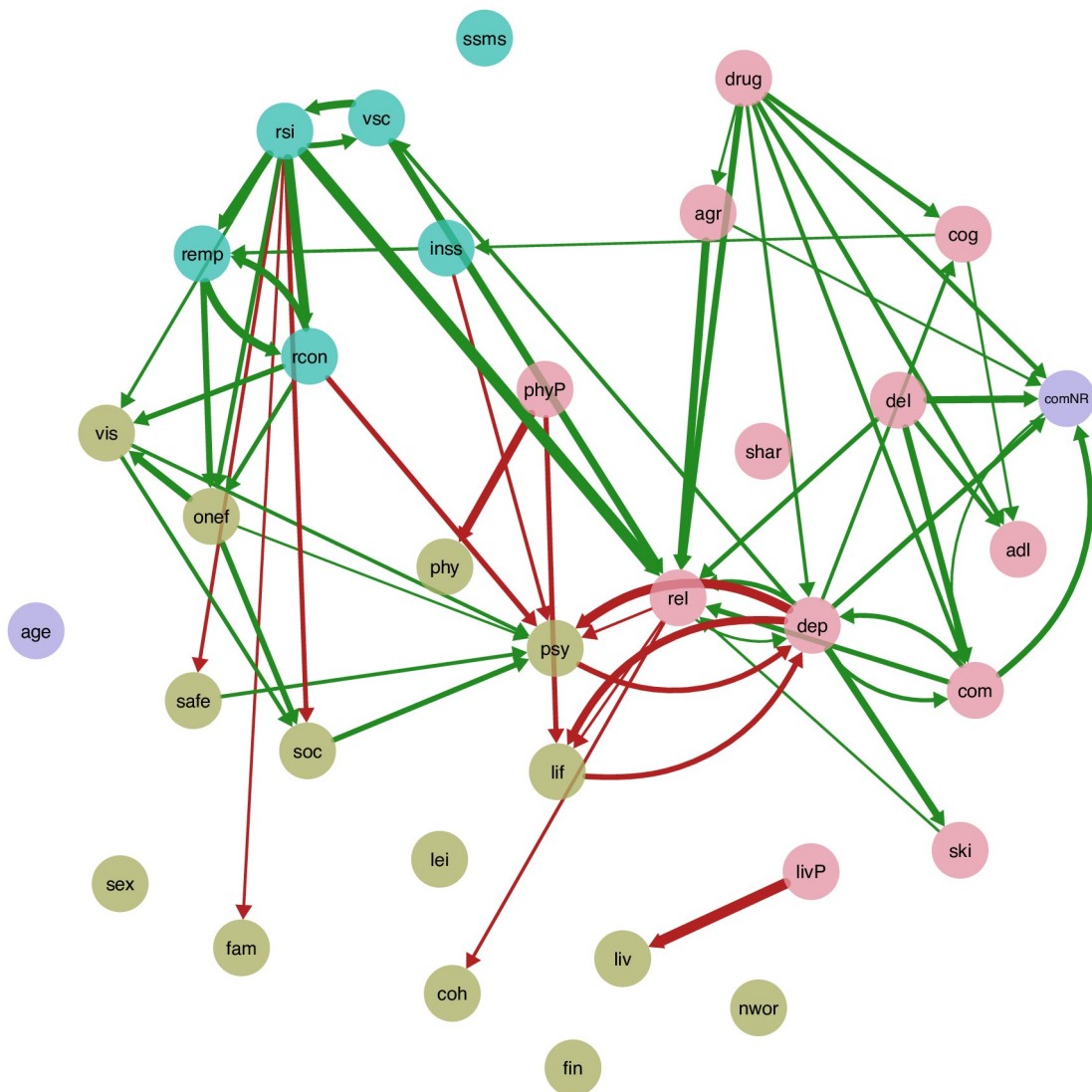

**Fig 1. Depiction of a previously published association network based on self-reported data from autistic adults.** This association network depicts the interrelations between ASD characteristics, well-being domains, and aspects of daily functioning as found in self-reported data presented in Deserno et al., 2017. The nodes represent the same variables as depicted in Fig 3, while the edges represent regularized partial correlations between those variables.

The MANSA has been shown useful to obtain accurate quality of life data [24]. The self-report questionnaire consists of 26 items covering 14 subjective well-being domains, such as general life satisfaction, social satisfaction, and satisfaction with personal safety. We included all 14 items of the MANSA in the PCR scale.

The HoNOS is a viable instrument designed to assess daily functioning in different domains. The questionnaire consists of 12 items covering, for example, behavioral problems, social problems and cognitive problems. We included all 12 items of the HoNOS in the PCR scale.

The association network derived from the Deserno et al. [17] study was based on regularized partial correlations, where edges depict unique relationships between sets of variables controlling for all other variables in the network ([26]; see [27] for an accessible overview). To

work with the exact same network elements as in the previous study, we constructed the PCR scale based on the 32 items mentioned above plus two additional variables: *age* and *number of co-occurring diagnoses*. Raters were asked cause-effect questions in regard to each item from the constructed PCR scale, e.g. "To what extent do you think depressed mood causes cognitive problems associated with ASD?" and, likewise, "To what extent do you think cognitive problems associated with ASD cause depressed mood?". For any given item pairing, participants rated the perceived causal association with response options from 0 to 10, with 0 and 10 denoting "Not at all" and "Strong cause", respectively. With this scaling methodology, one can gain insight into how clinical experts themselves perceive the causal organization of the given elements in a network.

Since all possible combinations of the 34 network elements would have resulted in 1122 cause-effect ratings, administering all item combinations to all clinicians proved infeasible. Therefore, we decided to split the association network in three (overlapping) parts with an (almost) equal number of nodes ($j_1 = 14$, $j_2 = 13$, $j_3 = 13$) based on their clustering in the association network to ensure study feasibility. That is, we grouped items optimizing three parameters: clustering (i.e. cutting through as few edges as possible), equal group size and overlapping nodes with high degree centrality. This resulted in three subsets of variables, each covering items from two of the three scales. Hence, we chose reliable ratings, i.e. large rater groups, of a subset of all possible cause-effect ratings above less reliable ratings, i.e. small rater groups, of the full network. This also allowed us to assess inter-rater reliability. Clinicians who agreed to participate were randomly assigned to one of these three parts ($N_1 = 10$, $N_2 = 10$, $N_3 = 9$) and were asked to complete the PCR scale based on their clinical experience. Afterwards, we combined the information given by the 29 different raters to partially reconstruct the network structure insofar as the design allows (see below). We conducted a follow-up assessment in which we asked clinicians who had participated in the first rating task to take a look at the constructed network, based on their averaged ratings, and choose three intervention targets from the complete list of nodes depicted in the network visualization. Note that by limiting the answers to three targets we tried to avoid complete rank-ordered lists of all factors. The number of *three* answers was arbitrary in itself, but a limited number of intervention target resembles a typically realistic intervention context.

## Statistical analysis

**(i) Perceived causal pathways in the well-being system.**   First, in order to explore the network of factors, we constructed a network model based on the information retrieved with the PCR scale from the raters in this study, see Fig 2. The perceived causal relations that the clinicians indicated were recorded and averaged to create an adjacency matrix, where each cell represents the averaged value attributed to the relation between any two factors. We merged the resulting matrices from the three rater groups that rated different (but partly overlapping) parts of the network to create a partial reconstruction of the association network based on self-reported data with the exact same nodes. Each value in the adjacency matrix represents the cause-effect rating from the PCR scale made by the raters, with all unrated factor pairings coded as missing. We constructed the directed network (i.e., perceived causal network) from this merged adjacency matrix using the R-package "qgraph" [26]. For reasons of clarity and comprehensibility, only those relations endorsed by the raters with an average rating of at least 6 (on a scale from 0 to 10) on the PCR scale were included in the visual representation of the network. Manually thresholding the visual representation was necessary since the raters tended to attribute very high values to edges they thought were present. Please note that we did not specify such thresholds for any of the analyses described below. We explored network

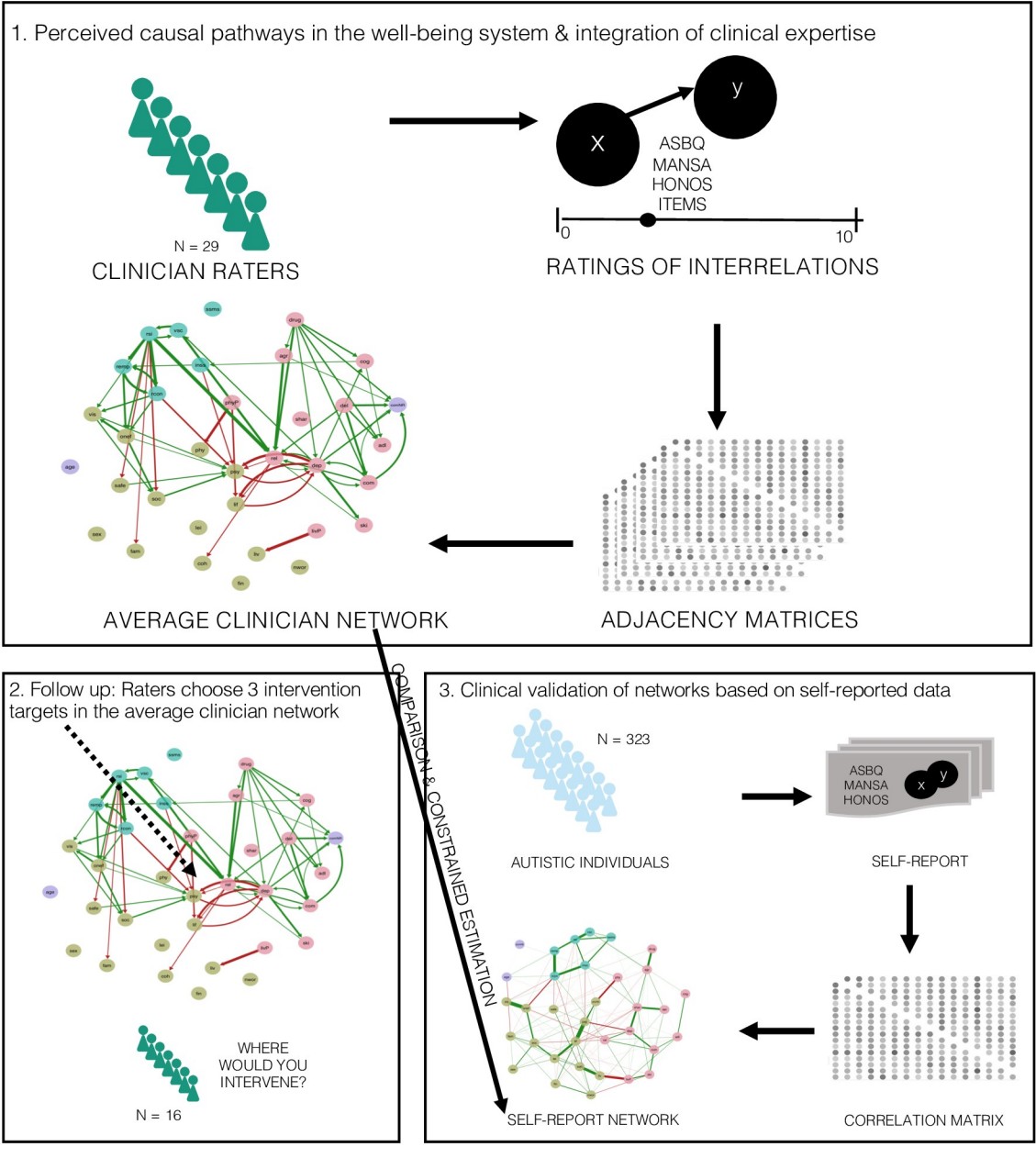

**Fig 2. Diagram of the three analytic steps of the current study.**

characteristics, such as degree centrality [28 - the number and strength of in- or out-going connections each factor has. Out-degree refers to the sum of the weights of the edges leaving a node, whereas In-degree reflects the sum of the edge weights of connections arriving at a node. In other words, we investigated what factors (i.e., nodes) are perceived to have a lot of causal influence (i.e., edges) on other factors and what factors are more often attributed as the effect (In-degree) versus cause (Out-degree) in all factor-pairings.

**(ii) Clinical validation of networks based on self-reported data.** Second, we took three different approaches in comparing the network structures found in the different PCR rater groups to the respective substructure of the association network from Deserno and colleagues

[17] shown in Fig 1. In a first step, we focussed on the global structure of these networks, ignoring the quantitative differences in edge weights. In this step, our main question was whether the edges that are present in the respective part of the PCR network are concordant with the association network based on self-reported data. To address this issue, we classified positive and negative edges according to the valence of the edge weight, i.e. the mean rating of that specific edge and calculated the proportion of edges that are concordantly classified as either positive, negative, or absent in both networks—compared to (i) all possible (present and absent) edges in the PCR network and (ii) compared to the edges *present* (both positive and negative) in the two networks. In a second step, we then focused on whether the edges that are present in both networks are similar in terms of weight. Because the PCR network is directed, while the association network is not, to be able to compare the two adjacency matrices, we averaged the unidirectional perceived causal relations between each two variables in the PCR network, resulting in a symmetric adjacency matrix with only *one* bidirectional coefficient for each pair of nodes; our justification for this procedure is that more directional effects should result in higher partial correlations as represented in the association network. To enable all readers to replicate the analysis, both adjacency matrices are provided as supplementary material online.

**(iii) Integration of clinical expertise into empirical network studies.** In a third step, we implemented a Bayesian framework for integrating a priori knowledge (here: the structure of the rated networks) in the estimation of an empirically derived network structure (here: the association network based on self-report data). Using the PCR framework as a scaffolding structure, we constrain the estimated association network based on self-report data (following [29]): those connections for which our raters indicate no evidence are forced to be absent in the estimated network. Simultaneously, the connections that correspond to a perceived causal relation are estimated from the data as usual. The practical implication of this is that our network consists only of the relations that we know a priori to be relevant. At the same time, this has methodological advantages as it reduces the number of free parameters to be estimated, which means fewer observations are required for accurate network estimation results. Because this integration requires a design in which ever edge is assessed by both of the relevant techniques, we could only apply this technique for the three subnetworks that had been both completely rated by experts and assessed empirically (see Methods section). In this constrained estimation approach, first the symmetric PCR adjacency matrices are used to define the probabilities of connections per node pair. Second, a thousand constrained networks are generated at random, using these symmetric PCR scores of each edge as its probability. Third, we use the R-package *BDgraph* to estimate the association network weights, given the provided structure. Fourth, we average the estimated networks across the generated samples, to come up with the Bayesian model average estimated network.

## Results

Most raters answered all questions, with only 10% of all PCR ratings missing. We did not replace these values and used all available data for the network analysis. We looked at the consistency of these ratings across raters: Cronbach's alpha indicated good inter-rater reliability within each group of raters, 0.85, $\alpha_2 = 0.80$, $\alpha_3 = 0.85$.

### (i) Perceived causal pathways in the well-being system

Many associations apparent in the association network, shown in Fig 1, remain in the PCR network. Fig 3 depicts the directed network based on the clinician ratings. Several features are notable.

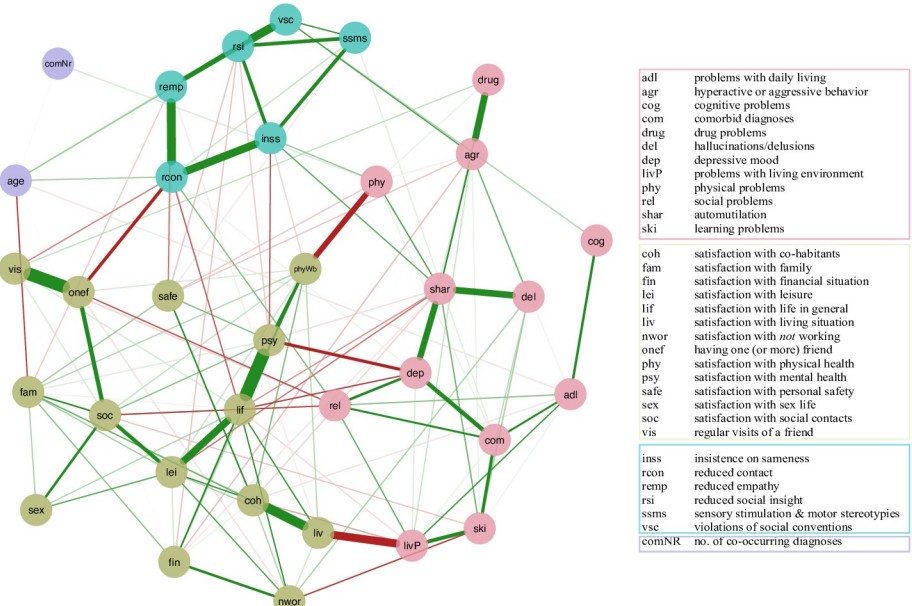

**Fig 3. This PCR network is a merged visualisation of three subnetworks based on ratings by three clinical expert groups and depicts the cause-effect ratings between ASD characteristics (blue), well-being domains (brown), and aspects of daily functioning (rose) as perceived by the clinicians.** The nodes each represent a unique variable and the edges represent the averaged directed cause-effect rating. For reasons of visual comprehensibility, we only included edges with an edge weight higher than 0.5 and used curved arrows in case of bidirectional relations.

First, the PCR network was highly connected. This means that clinicians indicate that there is a dense set of perceived causal effects involving the relevant variable, supporting the idea that they indeed form a causal network. Compared to previous PCR network studies [14, 15, 21], we found exceptionally few feedback loops. Only *reduced social insight* (rsi) and *violations of social conventions* (vsc); *reduced empathy* (remp) and *reduced social* contact; *the number of comorbid diagnoses* (comNR) and *comorbid problems* (com); *depressed mood* (dep) and *satisfaction with life in general* (lif); and *psychological well-being* and *depressed mood* (dep) were attributed bidirectional relationships.

Second, among all ASD symptoms included in the network, *reduced social insight* (rsi) ranked highest on both In- and Out-degree. In other words, clinicians rated *reduced social insight* more often as the cause of variance in other (central) factors, such as *problems with relationships* (rel), but also as the underlying cause of variance in (most) other ASD characteristics, i.e., *reduced empathy* (remp), *reduced social contact* (rcon), *violations of social conventions* (vsc) and *insistence on sameness* (inss). Also, one other ASD symptom ranked among the four variables highest on In- and Out-degree: *insistence on sameness* was attributed both many incoming and many outgoing connections, with for example a strong negative connection to *psychological well-being* (psy). From all ASD characteristics, *sensory stimulation/motor stereotypies* (ssms) was the one with least connections in the network as it was only rated as a cause for *problems associated with daily life* (adl).

Third, centrality indices (see Fig 4) showed *depressed mood* (dep) and *problems with relationships* (rel) as having the highest degree centrality amongst both well-being (brown) and daily functioning variables (rose). In addition, *depressed mood* ranked, together with *psychological well-being*, highest on betweenness centrality in the network, i.e., these nodes most often funnelled the shortest path between two other nodes in the network. Also, *depressed mood* was

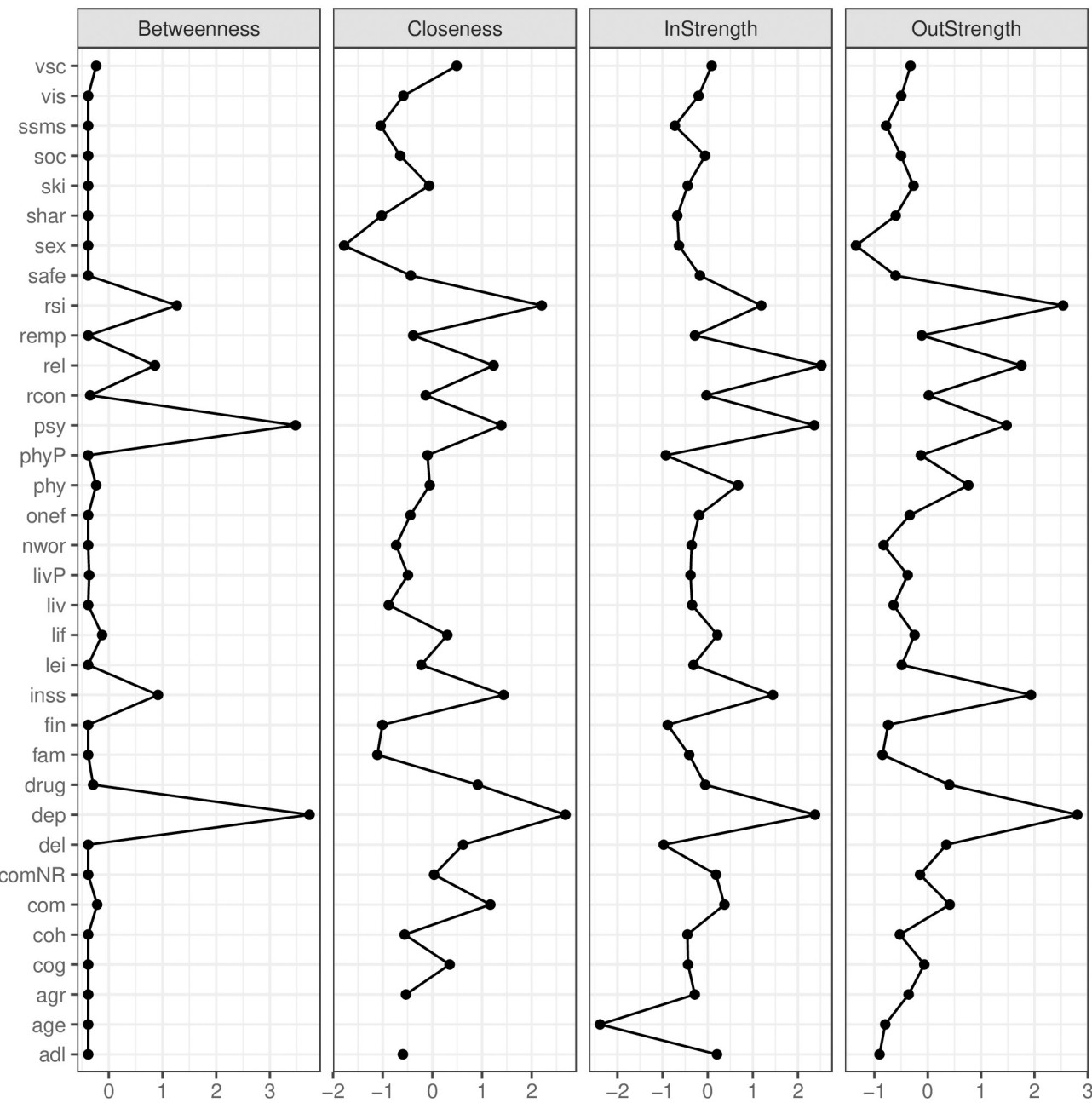

**Fig 4. Centrality indices for the PCR network depicted in Fig 3.** For the meaning of each node abbreviation see previous Figures and S1 Table.

the strongest predictor for both low *psychological well-being* and low *general life satisfaction* (lif), closely followed by *reduced social contact* (and *insistence on sameness* for *psy*) and *physical problems* (phy; for lif).

Finally, *age* and *sexual well-being* (sex) were attributed neither incoming nor outgoing connections. This means that our clinician sample did not consider these variables relevant causes of other variables in the network, and, in case of *sexual well-being*, a result of any other variable in the network (please note that we did not include any questions asking participants to rate

the effect of some variable in the network on *age*, because this would not result in meaningful questions, e.g. "How much do you think *depressed mood* causes *age*?").

Half of the clinician sample (N = 16) responded to our follow-up question, inquiring about *where* they would intervene in the provided network (Fig 3) if the goal would be to improve the general well-being of an autistic individual. For this follow-up question, we did not collect personal information. The question resulted in a broad spectrum of intervention choices: 21 out of 34 nodes were at least once chosen to be a target of intervention. The top four choices, however, were *depressed mood* (dep; 7 votes), *hyperactive or aggressive behavior* (agr; 5 votes) and *reduced social insight* (rsi; 4 votes) and *problems with drugs* (drug; 4 votes).

## (ii) Clinical validation of networks based on self-reported data

In this section, we compare the clinician network and association network based on two classifications that characterize the structure of the network: the presence/absence of edges and their valence (positive/negative).

To be able to zoom in on different parts of the networks, we depicted the relationship between the values of the PCR network and the association network in Fig 5, which presents a scatterplot of the edge weight values in the clinician and the association network and visualizes information on both the concordance in *structure* of the networks (whether an edge is present or absent in both networks) and the alignment of the relative strengths of edge weights in the networks. The values that form a line parallel to the x-axis represent edges that are 0 (absence of an edge) in the association network. The reason that there are no values parallel to the y-axis (absence of an edge in the clinician network) is that the averaged ratings never resulted in a value of exactly 0. The values in the lower left section of the scatterplot represent edges that are negative in both graphs (e.g., agreement in both networks that *psychological well-being* has a negative relation with *depressed mood*), while the values in the upper right section of the scatterplot represent edges that are positive in both graphs (e.g., agreement in both networks that *hyperactive or aggressive behaviour* has a positive relation with *drug problems*). The two values in the lower right section are edges that are positive in the clinician network and negative in the association network, while the value in the upper left section represents an edge that is negative in the clinician network and positive in the association network (i.e., the findings in the two networks are opposed to each other).

One possible metric to investigate the similarity of the networks' *structure* is to simply look at the proportion of edges that are concordantly classified as either positive, negative or absent in both networks. Since there we no edges in the clinician network with a value of exactly 0, we decided to look at the concordant classification of *present*, both positive and negative, edges in the two networks. Making this comparison, 96% of all edges present in the clinician network are matching in their classification as positive or negative when compared to the association network.

When comparing the *weighted* edges that are present in both networks, a first clear difference that can be inferred from the adjacency matrices is that the weights that result from the average ratings of the clinicians are higher (both negative and positive weights) than the regularized partial correlation values. This difference in magnitude of the weights should not be interpreted in substantive terms, as it follows simply from the different scale of the weights: weights in the association network reflect partial correlations scaled between -1 and 1, whereas the weights in the adjacency matrix of the clinicians result from averaging the ratings of clinicians on a scale from 0–10. For this reason, to compare the association network to the PCR network we consider only the *relative* magnitude of the weights. That is, are the edges that are relatively strong in one network also relatively strong in the other network?

## Scatterplot of Adjacency matrices

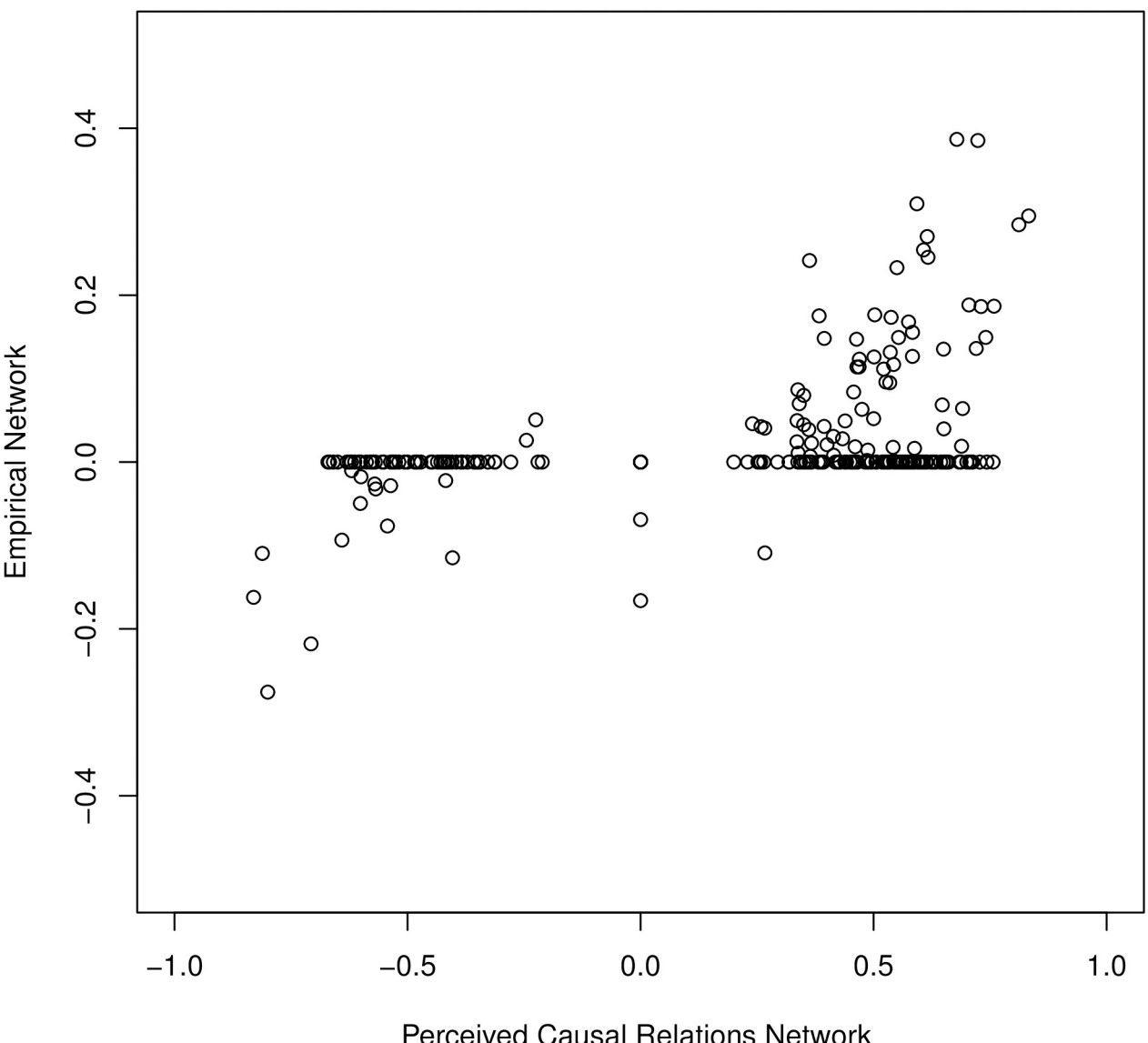

**Fig 5. This scatterplot depicts the relationship between the values of the PCR network and the association network.** The values on the line parallel to the x-axis represent absent edges (value of 0) in the association network. The values in the lower left section represent edges that are negative in both graphs, while those values in the upper right section represent edges that are positive in both graphs.

Because not all edges that are present in the association network have been rated by the clinicians, we calculated the correlation between the weights in both networks only for those edges that had actually been rated by the clinicians. This results in a correlation of $r = 0.43$. This correlation is likely to be a lower bound estimate for the correspondence between networks, because the graphical lasso indiscriminately sets edges to zero when they contribute insufficiently to model fit (this produces the horizontal pattern of points in Fig 5), while the PCR network provides continuous variation across all edge weights. This leads to restriction of range in the graphical lasso estimates, which is well known to attenuate the correlation

**Table 1. Contingency table for concordant and disconcordant positive (1), negative (-1) and absent (0) links in the PCR and the empirical network.**

| Empirical | | |
|---|---|---|
| PCR | -1 | 1 |
| -1 | 14 | 3 |
| 0 | 52 | 107 |
| 1 | 2 | 66 |

We could not include all possible links in the PCR scale, resulting in induced zeros in the adjacency matrix (i.e., absent edges, by design, in the PCR network). We did not include those absent edges in the contingency table nor the calculation of the reported metrics.

coefficient. Therefore, we also looked at the relationship between the edges that are *present* in *both* networks and as such cannot be affected by restriction of range (i.e., only those values that are located away from the horizontal axis corresponding to the value zero in Fig 5). This correlation coefficient equals $r = 0.73$, mainly driven by the fact that most positive edges in the association network were also attributed positive edge weights by the clinicians and most negative edges in the association network were attributed negative weights. When zooming in on the correlation of only the negative edges present in both networks (depicted in the lower left section of Fig 5) or only the positive edges present in both networks (depicted in the upper right section of Fig 5), the correlation coefficient equals 0.57 or 0.62, respectively, indicating moderate correspondence. This suggests that the PCR network and the association network align strongly in terms of the signs of relations between variables, moderately in terms of the magnitude of these relations, and feature a weak-to-moderate correspondence when the pattern of structural zeroes induced by the graphical lasso is not accommodated for, see Table 1.

## (iii) Integration of clinical expertise into empirical network studies

To integrate the clinician and the association network, we applied the constrained estimation approach described in Hinne et al. [29], i.e. the estimation of the ASD association network constrained by the information taken from the clinician network. Because this integration requires a design in which ever edge is assessed by both of the relevant techniques, we could only apply this technique for the three subnetworks that had been both completely rated by experts and assessed empirically (see Methods section). In this constrained estimation approach, first the symmetric PCR adjacency matrices are used to define the probabilities of connections per node pair. Second, a thousand constrained networks are generated at random, using these symmetric PCR scores of each edge as its probability. Third, we use the R-package *BDgraph* to estimate the association network weights, given the provided structure. Fourth, we average the estimated networks across the generated samples, to come up with the Bayesian model average estimated network.

Fig 6 shows the resulting networks, next to the networks as estimated by the graphical lasso for comparison. Qualitatively, these figures confirm that the perceived causal relation structure overlaps greatly with the networks obtained from questionnaires. Only a handful connections are absent in the constrained approach and present in the lasso estimates or vice versa. The connections with a difference larger than 0.1 are:

**Subgraph 1**

- from depressed mood to self-harming behavior

- from problems with delusions to self-harming behavior

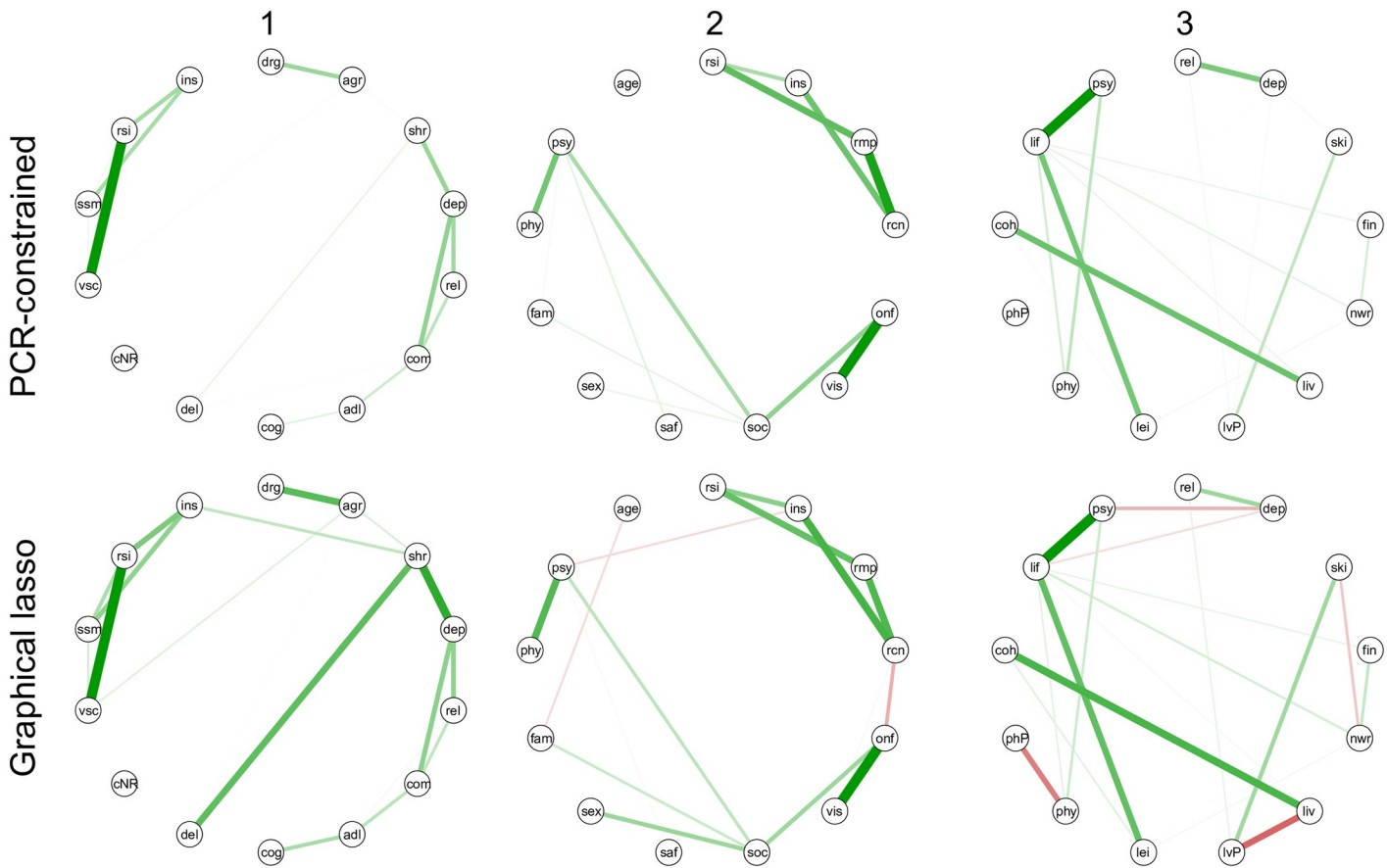

**Fig 6. In the first row the three subnetworks are estimated with constraints based on the network structure found in the clinician network (PCR-constrained).** In the second row, the graphical lasso networks are depicted individually, for comparison (Graphical lasso).

**Subgraph 2**

- from psychological well-being to insistence on sameness

- from having a friend to reduced contact

- from visiting friends to reduced contact

- from sexual well-being to social satisfaction

**Subgraph 3**

- from satisfaction with not working to relational problems

- from general satisfaction with life to depressed mood

- from psychological well-being to depressed mood

- from satisfaction with not working to problems with developing skills

- from problems with living situation to satisfaction about living situation

Interestingly, for the latter two subnetworks, those connections that are excluded by the PCR-constrained estimate, but are present in the graphical lasso estimate, correspond mostly with negative partial correlations.

## Discussion

### Main findings

The current network study is the first to compare how clinicians *think* that ASD symptoms and wellbeing are interrelated with how they are interrelated in self-report data based on self-reports by autistic individuals. Results suggest a moderate to strong alignment of the networks arrived at through both approaches, suggesting that clinician networks and association networks may, at least in part, point to the same underlying structure of potentially causal relations. The moderate convergence of these methods also suggests that promising insights are readily available through a synergy between PCR networks and association networks, which could capitalize of the strengths of both. This indicates that the integration of these techniques is a promising methodology that should be further studied, and we have provided the first workable solution to this challenge through the Bayesian constrained estimation approach. Finally, the current study provides the first validation study of widespread network estimation procedures (i.e., regularized network estimation; [27]) to use independent sources of data, by comparing it to PCR methodology to assess expert ratings of perceived causal connections. The alignment between the association network and the PCR methodology supports the validity of both methods, although research investigating sources of divergence is naturally called for as well.

### Limitations

When interpreting the similarities and differences of these networks, several limitations deserve mention. First, splitting up the network in three parts for reasons of feasibility resulted in leaving out many pairings of factors for the PCR scale. This means that, as mentioned in the Results section, not all connections present in the association network have been rated by the clinicians. When setting up the study, we chose large rater groups above a complete rating of the association network, which safeguards reliability of the estimated network structures, but limits the breadth of the investigation. Note that for rater- and structured interview studies a sample size of 29 raters is considered large [30]. Future studies, however, might focus on the latter if they aim to advance the thorough comparison of these two types of networks, or may attempt to develop optimal ways of distributing raters across parts of the network to optimize reliability and breadth jointly.

Second, it is important to note that the centrality of nodes in the association network indicates the importance of a node for the given network structure *within* the autistic population as it is mainly based on self-reported data of people with an ASD diagnosis. This means that, for example, suffering from depressed mood, which is highly central, impacts the state of all other factors in the network *for this specific* population (assuming that the variables in question indeed form a causal system with symmetric effects). It does not reveal information about whether this is a distinct feature of the network with respect to other populations. For example, when researching a network of factors in the general population and we look at what might be a central predictor for being a successful basketball player, the network is likely to reveal that someone's *height* plays an important, i.e., central, role. When looking at the same network of predictors in a sample of professional basketball players, their height will probably not be as central in this population-specific network as they do not vary much in terms of height. In the same vein, the PCR network might differ from the association network in terms of centrality

of certain nodes as clinicians might have rated all connections aiming to represent what nodes are important for the autistic population compared to other clinical populations or typically developing people.

Third, in this study we were able to reach out to highly experienced experts in the autism field: the knowledge that we combined into the PCR network was based on about 14 years, on average, of clinical work with people with an ASD diagnosis. This is a very specific sample of experts, of course, which limits the generalizability of the ratings to, for example, other mental health professionals or general practitioners [31]. Also, although we know that the majority works for clinical institutes that have a tradition for over 40 years in specialized autism teams, we did not specifically ask for more detailed information about their background and type of experience. This leaves us unable to assess whether the subgroup asked to choose intervention targets has very specific or, instead, a diverse range of characteristics. Another important factor regarding the generalizability of our results concerns the choice of what factors to include in the PCR rating task. In the current study, this choice was a priori limited by the available data that our association network was based on, i.e. the questionnaires that were implemented in the treatment monitoring systems of a Dutch autism clinic [17]. Relatedly, the association network in this study was based on the limited available data from one specific mental health clinic, resulting in an inability to verify exact IQ scores and lack of generalizability to those with intellectual disabilities.

## Relation to the literature

In this study, the PCR network revealed the causal model ASD clinicians adhere to when they think about the interrelatedness of ASD symptoms. Results suggest that, in the first place, *reduced social insight* is seen as the underlying *cause* of the other ASD symptoms in the network. Other than the influence of *reduced social insight*, the ASD symptoms are not attributed any strong incoming connections. Rather, they appear to be seen as *exogenous* variables, which cause individual differences in well-being and daily functioning, but are not themselves caused by the other variables in the system. This would be consistent with the plausible idea that ASD symptoms arise from sources external to the factors in the current networks, e.g., from problems associated with brain development. For example, early brain parameters are not assessed in the current study, but without any doubt relevant to atypical development (see [32] for an elaborate review of this line of thought). In addition, two pairs of ASD symptoms are connected by strong feedback loops (*reduced social insight* with *violations of social conventions*, *and reduced empathy* with *reduced social contact*) suggesting the plausible hypothesis that these problems mutually reinforce each other. A strong impact on domains of well-being and daily functioning, on the other hand, is also attributed to the ASD symptom *insistence on sameness*, which has perceived causal connections to e.g., *problems with daily functioning* and *problems with relationships*.

Second, the clinicians' choices regarding the factors that they would make their target of intervention (depression, reduced social insight, and aggressive behavior) were related to the respective centrality of these nodes in the PCR network. The factor that ranked highest among all choices was also the most central factor in the PCR network: depressed mood. This is in line with literature demonstrating that clinicians causal reasoning when dealing with diagnostic information concerning mental disorders is related to the causal model they adhere to [33–35].

Third, we found that the way this clinicians sample perceive cause-effect relations between ASD symptoms, well-being, and domains of daily functioning is fairly similar to the interrelatedness of these factors found in self-reported data. All links that are present in both the

association network from Deserno et al. [17] and the current PCR network, are attributed similarly weighted cause-effect ratings by the clinicians as their edge weights found in the self-reported data. At the same time, this suggests that clinicians are aware of the specific impact certain variables in the well-being network have on each other, and that the association network may pick up the relevant relations as well. This finding is concordant with earlier research, which has suggested that clinicians' personal cause-effect models affect their diagnoses [18] as well as their judgement of the effectiveness of a specific intervention [36–38]. The congruence of these cause-effect ratings with the relationships found in self-reported data suggests that there are no major gaps between the two concerning those relationships that are present in both networks. It is important to note, however, that not all possible edges in the well-being network were rated by the clinicians, so we are unable to assess to what extent this result generalizes to parts of the network that have not been rated.

## Outlook

In sum, we have presented a useful way of translating clinical expertise in the ASD realm into quantitative information and hereby illustrate a promising way to integrate clinicians' knowledge into scientific studies. Future studies could use these tools to quantify different types of knowledge. For example, as many voices have been campaigning for more participatory research in (not only) the ASD realm, the PCR methodology could be used to build new models and generate hypotheses in cooperation with groups of autistic people or any other knowledgeable informant. Advances in this methodology could even result in a tool worth implementing in treatment and diagnosis. The schematic representation of perceived cause-effect models might benefit both clinician and client in any mental health setting (see also [39]). In addition, we have provided the first validation of psychological network estimation procedures that have been energizing different clinical fields in psychological science. The combination of association networks and PCR ratings offers a promising framework to assess the validity of network structures found in self-reported data. However, in order to structurally compare these types of networks, it is important to develop advanced statistical techniques in future research. Here, we illustrate what important insights are to be gained into the interrelatedness of ASD and well-being by using the PCR methodology alongside self-reported data. We are convinced that the complementary use of quantified clinical expertise and self-reported data offers novel opportunities to study the workings of any multi-causal complexity in clinical psychology.

## Supporting information

**S1 Table. List of abbreviations of nodes in the networks.**
(DOCX)

**S1 Data.**
(CSV)

**S2 Data.**
(CSV)

## Author Contributions

**Conceptualization:** Marie K. Deserno, Denny Borsboom, Sander Begeer, Hilde M. Geurts.

**Data curation:** Marie K. Deserno, Hilde M. Geurts.

**Formal analysis:** Marie K. Deserno, Riet van Bork, Max Hinne.

**Funding acquisition:** Sander Begeer, Hilde M. Geurts.

**Methodology:** Marie K. Deserno, Denny Borsboom, Sander Begeer, Riet van Bork, Max Hinne, Hilde M. Geurts.

**Project administration:** Hilde M. Geurts.

**Resources:** Denny Borsboom, Hilde M. Geurts.

**Software:** Max Hinne.

**Supervision:** Denny Borsboom, Sander Begeer, Hilde M. Geurts.

**Writing – original draft:** Marie K. Deserno.

**Writing – review & editing:** Denny Borsboom, Sander Begeer, Hilde M. Geurts.

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
