## [Decision Letter · Decision Letter 0]

7 Aug 2020

PONE-D-20-15702

Highways to happiness for ASD adults? Perceived causal relations among clinicians

PLOS ONE

Dear Dr. Deserno,

Thank you for submitting your manuscript to PLOS ONE. After careful consideration, we feel that it has merit but does not fully meet PLOS ONE’s publication criteria as it currently stands. Therefore, we invite you to submit a revised version of the manuscript that addresses the points raised during the review process.

We look forward to receiving your revised manuscript.

Kind regards,

Geilson Lima Santana, M.D., Ph.D.

Academic Editor

PLOS ONE

Journal Requirements:

Reviewers' comments:

Reviewer's Responses to Questions

**Comments to the Author**

1. Is the manuscript technically sound, and do the data support the conclusions?

Reviewer #1: Partly

2. Has the statistical analysis been performed appropriately and rigorously? 

Reviewer #1: Yes

3. Have the authors made all data underlying the findings in their manuscript fully available?

Reviewer #1: Yes

4. Is the manuscript presented in an intelligible fashion and written in standard English?

Reviewer #1: No

5. Review Comments to the Author

Reviewer #1: Review PONE-D-20-15702

### GENERAL COMMENTS

In general, this is a very interesting manuscript that has used a network methodology to shed a light on a new methodology (PCR) to catch clinical aspects regarding ASD.

Nevertheless, to achieve the best shape, the manuscript needs some modifications.

The authors tend to use several personal choices to justify some decisions (without a proper rationale to justify them).

Two of the most problematic topics of the article are:

a) the sample size (n=29 completed the questionnaire and n=16 were enrolled in the follow-up),

and b) is the use of the term "causal". If by one side, the authors mention correctly "association network", by the other, they tend to classify associative relationships as causal ones. The authors should avoid using “causal” when data is correlational.

The authors should have more care regarding the organization of the text. For example: the tables are out of order, the methodology and discussion presents redundant parts, and more than once, the reader needs to read again a part of the text to understand the intentions of the authors.

Rather than using the term "empirical data", it would be more appropriate to rename this sort of information as "subjective data" (gathered directly from patients) in opposition to "objective data" (gathered from clinicians regarding their patients). Because, as a matter of fact, this study compares personal observation of patients with observations from clinicians. Since "empirical data are facts that are obtained by observation or experiment", both data used in the research could be considered empirical. The definition used by the authors does not consider as empirical the second cluster of data. Nevertheless, this assumption is not correct.

In a more general comment, to improve the comprehensibility of the text, it would be very interesting if the text was built having blocks of information in mind. Them, pairing each part of the text with the next section.

Example: The concepts and gaps mentioned in the introduction should be paired with the aims of the study. Then, these previous parts being paired with the corresponding part on the methods used for answering each of these questions (in a step-by-step fashion). Followed by the results you have for each of those questions. And, finally, the discussion of these results (still pairing them with the previous blocks), and the presentation of a conclusion

### INTRODUCTION

• LINES 65 to 68 – I suggest the authors place the primary purpose of the study at the end of the introductory section. The same could be said regarding information regarding LINES 90 to 93.

• LINES 85 to 93 – The authors should think about dividing the sentence in two at least. Since it is too long, it is difficult to follow, and the reader has to read again to catch all the information it provides.

• LINES 94, 106, and 119 – The authors should reconsider the use of these numbered steps ("First", "Second", "Third"). It is unnecessary. Moreover, the paragraphs they begin with were the continuity of the literature review on the field brought by the introductory section. So, it doesn't make sense the use of them…

• LINES 108 and 109 – The authors make an imperative affirmation. Nevertheless, it is a supposition. So that they should use a conditional verbal form instead of such an illation.

— "This suggests that professionals working in the clinical field already WOULD/COULD/MAY conceptualize psychological constructs as a set of causal relations between symptoms and other factors"

• LINE 110 – It is not adequate mention "in our view". If the authors made a brief review and from that, they concluded that the sort of information is underused, so mention that. Avoid this sort of postulation. E.g.: "A BRIEF SEARCH IN PUBMED DEMONSTRATES THAT THERE IS LACK OF STUDIES…"

• In general, it is interesting the use of questions to incite the readers toward some aspects of the literature gap. Nevertheless, some of them could be placed as affirmations (and their respective references). The overuse of the questions may give the impression that the authors did not perform a good literature review (what doesn't seem to be the case). But to prevent this imprecise impression, use the questions only for the most important points raised in the text (and only those the authors intend to answer with the present manuscript).

In LINE 133-134, for example, the authors mention: "The aim of the current study is to take the first steps towards addressing these questions…". The problem is: there were so many questions… which one of them do they intend to answer? All of them?

• LINE 125 – Mention, in an apposition, what is Delphi Methodology (e.g.: "a structured communication technique")

• LINE 136 to 137: Divide the aims of the study. The too-long sentence could be better explained as: "The aims are: a)……… , b)………… , c) ………" . This separation will make the text easier for readers.

• The PCR methodology should be explained before the presentation of the aims/objectives (that should be the very last points of the introductory section).

• LINE 147 – I suggest changing the term "client" for "patient"…

• LINE 150 – Suppress "exploratory".

• LINE 151 – Although the proposal of look into Deserno et al. 2017 is interesting, that proposition would be more suitable for the reader if the article would be presented in a medium with hypertextual tools (which is not the case of PLOS ONE). So, explain very briefly, in the introduction, before presenting the aims, the domains of well-being as presented in Deserno et al., 2017.

• LINES 155 – 156 – Why do the authors would like to explicitly mention the PCR as an objective of the paper, deviating from the core results? ("…provide an example of how to use the PCR toolbox to integrate the knowledge of clinicians in empirical studies"). Is it necessary? Wouldn't this interesting work itself be a proper example?

• LINES 156 – 158 – More suitable for the "Methods section".

• LINES 159 – 167 – This paragraph would be unnecessary if the introductory section would be properly constructed. This last paragraph should be dedicated to the aims and objectives (that — in this article — were separated through the introduction. Moreover, sentences such as (LINES 164 – 165): "…the exact same variables found in Deserno et al. (2017)" should be avoided because the reader should have all the essential information for understanding this article without the necessity of reading additional articles. The last sentence should be placed in Methods.

### METHODS

• In general, this section is prolix, presents redundant parts, and is not straightforward. It needs to be reformulated to provide to the reader a better understanding of the step-by-step of what was done.

• Regarding the PCR network, remember always use "'perceived causal' effect" instead of "causal effect", "causal network", "causal relation" etc (ex.; line 270, 272) since one perception does not indicate a real causal effect.

• Concerning the sample, how many clinicians were invited to participate, and how was the acceptance rate among all clinicians invited? How the authors calculated (and eventually decided) that the number of raters was enough (n=29)? How do the authors think about the number of responders in the follow-up phase?

• It is not clear how the selection of the institutions and the raters was done. Do the services investigated were the only facilities treating ASD in the region? Provide a glimpse both the selection of the sample. The simple mention that the authors "knew all the clinicians" is not descriptive enough and does not allow replications. Provide objective criteria.

• Why the authors have included raters with different backgrounds (psychologists, psychiatrists, behavioral scientists, and social workers)? Since their formation is quite different from each other some bias on the appreciation of causal relations could certainly occur. Moreover, the PCR intends to access causal relations based on their clinical expertise. Even though the patients are the same, the clinical perception of a professional is biased by people's academic background.

• In the same direction, mention that because they work in such facilities, "we know that all clinicians participating in the current study are involved in diagnostic work, consultation, and intervention services" is not informative enough. Besides that, how the authors guarantee that some of the respondents do not work only in the administration or coordination of such settings rather than as clinicians.

• The authors should avoid expressions such as: "[they] consider 'advanced experience with ASD'". Such subjective evaluation weakens the methodology of the study. For example, mention only that "we knew that they are [specialists]" is too fragile. Objective criteria would be preferable.

• What is CASS18+? I suppose it is an institution/facility for the treatment of ASD. Explain previously used abbreviations.

• Provide a table with the demographic characteristics of the group.

• Provide to the reader a brief explanation of Deserno et al., 2017 and also Frewen et al., 2012 study. Despite hypertextual material is interesting, this sort of tool is not suitable for PLOS ONE and burdens the reader.

• LINES 204 to 210: That is "Statistical Analysis" rather than "Measure and Procedure".

• FIGURE 1 and 2: The quality of the images does not allow us to read the texts inside the nodes or in the figure legends. Figure 2 needs a legend.

• LINES 243 – 247: The authors mention that: "Therefore, we decided to split the EMPIRICAL network in three (overlapping) parts with an (almost) equal number of nodes (j1= 14, j2=13, j3=13) based on their clustering in the EMPIRICAL network to ensure study feasibility."

Is that correct?

• LINES 243 – 252: This was the less comprehensible part of the methodology. Reformulate it to allow a better understanding. It would help a lot providing a diagram/draw/scheme with the step-by-step of the methodology.

• LINES 250 – 252: "Large rater groups"? The author assessed only 29 raters… It is not clear what was done.

• The low number of raters is a serious question. And, when allocated to one of the three different groups, the number is even smaller (n=9, 10, and 10).

• LINE 259 – Why the authors have asked the raters to choose THREE targets? Based on what rationale?

• LINE 276 – Why the authors have used a threshold of 6? Based on what rationale?

• It also is not clear why, having in mind that the empirical network reflects partial correlations scaled between -1 and 1, why the authors have used clinicians' ratings from 0 to 10. Why not using a Likert scale, for example? Moreover, comparing two different networks with two different methodologies would not be problematic?

• I wonder if the authors could have further commentaries on the following sentence from LINES 276 and 277:

—"Manually thresholding the visual representation was necessary since the raters tended to attribute very high values to edges, they thought were present."

Do the authors did not expect that ceiling effect (since they search for "perceived causal effects", those how seem to the raters more intense would be valued)? Do the way the questions were asked did not favor that behavior?

Moreover, in LINE 278, the authors mention that "did not specify such thresholds for any of the analyses". Nevertheless, they mention that (LINE 275-275) "only those causal relations endorsed by the raters with an average rating of at least 6 (on a scale from 0 to 10) on the PCR were included in the visual representation of the network". It seems that one sentence contradicts the other.

### RESULTS

• The results section is difficult to follow and, sometimes, no distinguished from what should be in the discussion. The sentences are very long, and the excessive number of examples and comparisons with previous studies (that should be done in Discussion rather than in Results — E.g. LINE 339 – 343) makes it difficult for the reader to retain the main findings among the profusion of information.

The text would benefit a lot if the authors would be more direct in reporting the results, commenting them only at the Discussion. Focus on the main findings. Although this sort of methodology provides uncountable information, is detrimental to the text if the authors try to mention everything they find.

• The authors do not present numerical results in this section. This data is important and not presented only as supplementary material (LINE 359).

• All tables and figures must be close to their title, footnotes, and all material referent to them. Is difficult to follow the results because the figures do not present this information. Moreover, the Figures do not have titles, but long explanations of what they represent. That fact indicates clearly that the figures are not self-explanatory as they supposed to be (at least in partly).

• The quality of the images is poor making it impossible to read them.

• LINE 337 – The authors should provide the list of abbreviations in the text and not only in the supplementary material.

• The comprehension of the Tables and Figures should be independent of the text (and vice-versa). They have to contain all the necessary information for their comprehension (e.g. LINE 374 — the table title contains references to the text).

• There is no reference in the text regarding where Table 1 should be placed. It was put in the text but not mentioned.

• LINE 377 – 384: It is not clear why the authors decided to investigate the interventions the raters would do. That's not one of the aims of the study and with such a small number of raters, the results are not meaningful.

• LINE 387 – 393 and 458 – 469: This paragraph is not part of the Results section. It describes the methodology.

• LINE 431: "…between -1 AND 1…"

• Where is Figure 4 that is mentioned in the text?

• LINE 496 – 499: This paragraph is not part of the Results section and should be placed in the Discussion section.

### DISCUSSION

• In general, the Discussion is easier to read than the rest of the text, although repetitive.

Nevertheless, the authors tend to stretch the line in some conclusions to explain some findings. For example, I did not understand the sentence in LINE 526-528: "This would be consistent with the plausible idea that ASD symptoms arise from sources external to the current networks (e.g., from problems associated with brain development)."

How's that, "external to the current network"? Are not all the ASD symptoms part of a neurodevelopmental problem?

Moreover, having assessed only about 30 clinicians from a limited number of facilities is too forced, and — why not say — presumptuous, mention that "we found that the way clinicians perceive cause-effect relations between ASD symptoms".

• Again, in Discussion, the problem of "causal" inferences persists.

• I suggest the following articles in order to improve the quality of the text:

A. Docherty M and Smith R. The case for structuring the discussion of scientific papers. BMJ 1999;318;1224-1225;

B. Horton R. The hidden research paper. ‭JAMA,‭ June‭ ‬5,‭ ‬2002‭—‬Vol‭ ‬287,‭ ‬No.‭ ‬21‬‬.‬‬‬‬‬‬‬‬‬‬‬‬‬‬‬‬‬‬‬‬‬‬‬‬‬‬‬‬‬‬‬‬‬‬‬‬‬‬‬‬

• LINE 523: Wouldn't be "…other than THE influence…" rather than "…other than THAT influence…"?

• LINE 562: I do not think is fair the authors mention that they have used "large rater groups". The sample size is quite small and its subdivision smaller. This should be mentioned as a limitation.

• LINE 568: It is interesting the authors mention that "it is important to note that the centrality of nodes in the empirical network indicates the importance of a node…", but they relegate centrality measures to supplementary material.

• LINE 572 – 573: This is one of the most important sentences in the manuscript that deserves more attention from the authors:

— "…assuming that the variables in question indeed form a causal system with symmetric effects…".

It is important to mention that, based on an assumption that variables investigated to form a causal system according to the current literature, the networks were built. So, it is necessary to caution regarding the conclusions of the study.

• LINE 608: "schematic representation of PERCEPTION OF cause-effect " rather than "schematic representation of cause-effect "

6. PLOS authors have the option to publish the peer review history of their article (what does this mean?). If published, this will include your full peer review and any attached files.

Reviewer #1: No

---

## [Author Response · Author response to Decision Letter 0]

16 Oct 2020

Reviewer #1: Review PONE-D-20-15702

### GENERAL COMMENTS

In general, this is a very interesting manuscript that has used a network methodology to shed a light on a new methodology (PCR) to catch clinical aspects regarding ASD. Nevertheless, to achieve the best shape, the manuscript needs some modifications.

We would like to thank the reviewer for appreciating this manuscript and for suggesting modifications to achieve its best shape. We respond to all points being raised below and have addressed all helpful comments in the revision. We are grateful for the time investment of the reviewer and are convinced that the opportunity to revise the manuscript based on the reviewer’s suggestions has significantly improved our submission. 

1. The authors tend to use several personal choices to justify some decisions (without a proper rationale to justify them). Two of the most problematic topics of the article are:

a) the sample size (n=29 completed the questionnaire and n=16 were enrolled in the follow-up),

We respond to this comment below, comment no. 23, where the reviewer explained their concern more extensively. In short, for a rater study, based on the literature of structured interviews, the current study‘s sample size can be considered large (de Villiers et al., 2005). While we are of course aware that this is considered small for empirical studies based on survey data. We have now clarified this with the addition of lines 570 -573.

1. and b) is the use of the term "causal". If by one side, the authors mention correctly "association network", by the other, they tend to classify associative relationships as causal ones. The authors should avoid using “causal” when data is correlational.

We agree with the reviewer that careful use of the term ‘causal‘ is warranted. We have therefore revisited every single use of the word throughout the text and have either removed the word to avoid confusion or added the word “perceived” to stress that the discussed association is the results of a rating technique focused on perceived causality (the Perceived Causal Relation method).

2. The authors should have more care regarding the organization of the text. For example: the tables are out of order, the methodology and discussion presents redundant parts, and more than once, the reader needs to read again a part of the text to understand the intentions of the authors.

We thank the reviewer for spotting these inconsistencies and have carefully re-organized the text and tables to make sure there are no redundant parts anymore and to generally improve its comprehensibility. 

3. Rather than using the term "empirical data", it would be more appropriate to rename this sort of information as "subjective data" (gathered directly from patients) in opposition to "objective data" (gathered from clinicians regarding their patients). Because, as a matter of fact, this study compares personal observation of patients with observations from clinicians. Since "empirical data are facts that are obtained by observation or experiment", both data used in the research could be considered empirical. The definition used by the authors does not consider as empirical the second cluster of data. Nevertheless, this assumption is not correct.

We thank the reviewer for this suggestion and have given it extensive thought. In this context, the notion of subjective vs. objective may come with problematic implications: the clinicians’ data (now labeled as such, or as raters data) is also subjective in nature, as it is based on individual perception. At the same time, we agree that a pool of experienced clinicians as information source will be more objective than self-reported data as their judgments are based on different cases and years of experiences with autistic individuals. Based on this line of thought, we have, however, decided to refrain from using the ‘subjective‘ vs. ‘objective‘ phrasing but instead changed the ‘empirical‘ label throughout the data description into ‘self-reported data‘ and used the term ‘association network‘ instead of ‘empirical network‘. We hope to have accommodated the reviewers concern with these changes. 

5. In a more general comment, to improve the comprehensibility of the text, it would be very interesting if the text was built having blocks of information in mind. Them, pairing each part of the text with the next section.

Example: The concepts and gaps mentioned in the introduction should be paired with the aims of the study. Then, these previous parts being paired with the corresponding part on the methods used for answering each of these questions (in a step-by-step fashion). Followed by the results you have for each of those questions. And, finally, the discussion of these results (still pairing them with the previous blocks), and the presentation of a conclusion

We would like to thank the reviewer for this excellent suggestion to improve the comprehensibility of the text. We were trying to structure the introduction with first/second/third before but are much happier with this content-related step-by-step fashion. We chose the following headlines for the three aims of the current study (i) Perceived causal pathways in the well-being system, (ii) Clinical validation of networks based on self-report data, and (iii) Integration of clinical expertise into empirical network studies. We have implemented this structure throughout the whole manuscript, i.e. the introduction, the statistical analysis part of the method section, and the results. For the discussion section we decided to stick to the structure of integrating the reported findings in the broader context of the theoretical background. We have rewritten this section to make it more concise. 

### INTRODUCTION

6. LINES 65 to 68 – I suggest the authors place the primary purpose of the study at the end of the introductory section. The same could be said regarding information regarding LINES 90 to 93.

We thank the reviewer for this suggestion and have moved lines 65-68 and lines 90-93 accordingly. 

7. LINES 85 to 93 – The authors should think about dividing the sentence in two at least. Since it is too long, it is difficult to follow, and the reader has to read again to catch all the information it provides.

We thank the reviewer for this suggestion and have divided the sentence into two parts. We hope this improves its clarity: “Due to the explorative character of network analysis, it is unclear whether relations between variables identified in these studies actually reflect causal interactions (rather than, e.g., the effect of unmeasured common causes). In addition, it remains unclear what the direction of these causal pathways is, and to what extent the relevant pathways are also identifiable by other, independent modes of observation.”

8. LINES 94, 106, and 119 – The authors should reconsider the use of these numbered steps ("First", "Second", "Third"). It is unnecessary. Moreover, the paragraphs they begin with were the continuity of the literature review on the field brought by the introductory section. So, it doesn't make sense the use of them…

We thank the reviewer for pointing this out and have changed this to the (previously suggested) subheadings listed above, see our response to comment no. 5.

9. LINES 108 and 109 – The authors make an imperative affirmation. Nevertheless, it is a supposition. So that they should use a conditional verbal form instead of such an illation.

— "This suggests that professionals working in the clinical field already WOULD/COULD/MAY conceptualize psychological constructs as a set of causal relations between symptoms and other factors"

We thank the reviewer for this suggestion and have changed this into a conditional verbal form: “This suggests that professionals working in the clinical field may already conceptualize psychological constructs as a set of causal relations between symptoms and other factors.”

10. LINE 110 – It is not adequate mention "in our view". If the authors made a brief review and from that, they concluded that the sort of information is underused, so mention that. Avoid this sort of postulation. E.g.: "A BRIEF SEARCH IN PUBMED DEMONSTRATES THAT THERE IS LACK OF STUDIES…"

We thank the reviewer for this suggestion and have changed our wording accordingly: ‘A brief review of the existing network literature shows that this source of information has not yet been integrated in network approaches. It remains unclear, for instance, whether the network structures shown in self-reported data actually resemble those networks that clinicians would report.‘ (line 112-114)

11. In general, it is interesting the use of questions to incite the readers toward some aspects of the literature gap. Nevertheless, some of them could be placed as affirmations (and their respective references). The overuse of the questions may give the impression that the authors did not perform a good literature review (what doesn't seem to be the case). But to prevent this imprecise impression, use the questions only for the most important points raised in the text (and only those the authors intend to answer with the present manuscript).

In LINE 133-134, for example, the authors mention: "The aim of the current study is to take the first steps towards addressing these questions…". The problem is: there were so many questions… which one of them do they intend to answer? All of them?

We thank the reviewer for this comment and we agree that the overuse of questions can give the wrong impression. We have, therefore, rephrased some of the questions in the introduction into affirmations, see e.g. line 103-105: ' That is, the field is in dire need for a toolbox that can help us determine which connections in the network represent directed causal effects that arise from reciprocal causation or coupled equilibria, and which associations are due to the effect of unmeasured variables.' In (now) lines 129-134 however, we raise two very specific questions that we intend to answer with the current study: a) how to meaningfully represent clinical expert knowledge in network studies and b) how to combine existing methods assessing such expert knowledge with networks of self-reported data. Directly after raising these two questions, we provide a detailed explanation of how we plan to answer these questions by the current study. 

12. LINE 125 – Mention, in an apposition, what is Delphi Methodology (e.g.: "a structured communication technique")

We thank the reviewer for this suggestion and have added this information: “Recent studies, for example, have used the Delphi methodology (a structured interview/communication technique) to investigate the array of clinical practices used in the ASD realm (Wainer et al., 2017; Kerns et al., 2018).” (line 129)

13. LINE 136 to 137: Divide the aims of the study. The too-long sentence could be better explained as: "The aims are: a)……… , b)………… , c) ………" . This separation will make the text easier for readers.

We thank the reviewer for this suggestion and have divided the too-long sentence presenting the current study‘s aims to make it easier to read: “The first aim of the current study is to take the first steps towards addressing these questions, by constructing a symptom network on the basis of expert judgments (Frewen et al., 2012) to visualize the relationships among characteristics of ASD and multiple facets of outcome and well-being. The second aim is to combine this information with the information obtained from statistical analyses of survey data.”

14. LINE 147 – I suggest changing the term "client" for "patient"…

We thank the reviewer for this suggestion, and have changed our wording accordingly. 

15. LINE 150 – Suppress "exploratory".

We deem it important to clearly stress the exploratory character of this study. Since the reviewer did not share their reasons for this comment, we have, for now, decided to refrain from suppressing ‘exploratory'. 

16. LINE 151 – Although the proposal of look into Deserno et al. 2017 is interesting, that proposition would be more suitable for the reader if the article would be presented in a medium with hypertextual tools (which is not the case of PLOS ONE). So, explain very briefly, in the introduction, before presenting the aims, the domains of well-being as presented in Deserno et al., 2017.

We agree with the reviewer that it comes with disadvantages for the text‘s general comprehensibility to work with important information that is drawn from another article. We have, therefore, followed the reviewers‘ suggestion to add information on the ASD factors important to well-being from Deserno et al., 2017, see line 116-118.

17. LINES 155 – 156 – Why do the authors would like to explicitly mention the PCR as an objective of the paper, deviating from the core results? ("…provide an example of how to use the PCR toolbox to integrate the knowledge of clinicians in empirical studies"). Is it necessary? Wouldn't this interesting work itself be a proper example?

We thank the reviewer for this comment and have rephrased this aim without a focus on the PCR methodology: ‘...provide an example of how to integrate the knowledge of clinicians in empirical studies.‘

18. LINES 156 – 158 – More suitable for the "Methods section".

We thank the reviewer for this suggestion and have moved the sentence to the Methods section, lines 163 - 165. 

19. LINES 159 – 167 – This paragraph would be unnecessary if the introductory section would be properly constructed. This last paragraph should be dedicated to the aims and objectives (that — in this article — were separated through the introduction. Moreover, sentences such as (LINES 164 – 165): "…the exact same variables found in Deserno et al. (2017)" should be avoided because the reader should have all the essential information for understanding this article without the necessity of reading additional articles. The last sentence should be placed in Methods.

We agree with the reviewer and have restructured the introductory section, by moving parts to the Methods section, and removing redundant paragraphs. 

### METHODS

20. In general, this section is prolix, presents redundant parts, and is not straightforward. It needs to be reformulated to provide to the reader a better understanding of the step-by-step of what was done.

21. Regarding the PCR network, remember always use "'perceived causal' effect" instead of "causal effect", "causal network", "causal relation" etc (ex.; line 270, 272) since one perception does not indicate a real causal effect.

We have revisited every single use of the term ‘causal‘ to make sure it is clear that we talk about perceived causality, see our response to the reviewer comment no. 2.

22. Concerning the sample, how many clinicians were invited to participate, and how was the acceptance rate among all clinicians invited? How the authors calculated (and eventually decided) that the number of raters was enough (n=29)? How do the authors think about the number of responders in the follow-up phase?

As we sent out the survey through mailing lists of the clinical departments of the three mentioned institutions, we do not know how many clinicians have read the e-mail and decided not to respond. However, please note that whereas N=29 is a small sample size for quantitative statistics based on survey data (where each case represents one individual) the current sample size is particularly large for rater studies. Common rater samples often consist of less than 10 raters. 

We chose to select a very specific sample since we wanted to select a group of experts in the autism field based on their extensive experience. The follow up intervention ratings were obtained from a subgroup of 16 participants as only 16 of 29 clinicians responded to this second online assessment (although we sent it to all 29 clinicians with multiple reminders). We regret to report that we are unable to assess more specific characteristics of those 13 clinicians not completing the follow up assessment regarding intervention targets–and how this affecting the results. We have added a more explicit comment on the potential bias due to attrition to the discussion, lines 595–597 (‘This leaves us unable to assess whether the subgroup asked to choose intervention targets has very specific or, instead, a diverse range of characteristics.‘). 

We mention the limitations related to limited information on sample in lines 589 – 596: “Third, in this study we were able to reach out to highly experienced experts in the autism field: the knowledge that we combined into the PCR network was based on about 14 years, on average, of clinical work with people with an ASD diagnosis. This is a very specific sample of experts, of course, which limits the generalizability of the ratings to, for example, other mental health professionals or general practitioners (Nicolaidis et al., 2015). Also, although we know that the majority of our informants works for clinical institutes that have a tradition for over 40 years in specialized autism teams, we did not specifically ask for more detailed information about their background and type of experience.”

23. It is not clear how the selection of the institutions and the raters was done. Do the services investigated were the only facilities treating ASD in the region? Provide a glimpse both the selection of the sample. The simple mention that the authors "knew all the clinicians" is not descriptive enough and does not allow replications. Provide objective criteria.

Why the authors have included raters with different backgrounds (psychologists, psychiatrists, behavioral scientists, and social workers)? Since their formation is quite different from each other some bias on the appreciation of causal relations could certainly occur. Moreover, the PCR intends to access causal relations based on their clinical expertise. Even though the patients are the same, the clinical perception of a professional is biased by people's academic background.

In the same direction, mention that because they work in such facilities, "we know that all clinicians participating in the current study are involved in diagnostic work, consultation, and intervention services" is not informative enough. Besides that, how the authors guarantee that some of the respondents do not work only in the administration or coordination of such settings rather than as clinicians.

We distributed the questionnaire through the three leading clinical networks within the autism realm in the Netherlands, targeting specific individuals of which we knew that they are clinically speaking the authority in the Netherlands. Their affiliation with those institutions means that all clinicians participating in the current study are involved in diagnostic work, consultation, and intervention services - which is also the reason we selected these three leading clinical networks. Also, it is a Dutch standard that these institutes all work with multidisciplinary teams. We added this information in lines 167-176 of the manuscript. In addition, we assessed how many hours per week they engaged in clinical work and have added this information on page 13 of the revised manuscript. Although we know that the majority works for a clinical institute that have a tradition for over 40 years in specialized autism teams, we did not specifically ask for more detailed information about their background and type of experience. We, therefore, agree with the reviewer that there are some limitations due to the sample of the current study. We explicitly mention those limitations in line 589 - 598. Although individual differences related to professional background would be an interesting future research avenue, our intention was to find a way to find a meaningful way to represent clinical expert knowledge in network studies. For this purpose, it is beneficial to have a sample with a diverse background as this represents the range of expertise usually present at a clinical ‘case conference’, in Dutch clinics.

24.The authors should avoid expressions such as: "[they] consider 'advanced experience with ASD'". Such subjective evaluation weakens the methodology of the study. For example, mention only that "we knew that they are [specialists]" is too fragile. Objective criteria would be preferable.

We thank the reviewer for this comment. We consider these clinicians advanced, as the average clinical experience in the field was 14 years. We have now changed our wording to ‘years of experience’ in order to avoid subjective evaluations implied by ‘advanced‘, see lines 167.

25. What is CASS18+? I suppose it is an institution/facility for the treatment of ASD. Explain previously used abbreviations.

We thank the reviewer for pointing us to this missing definition, and have added it to the Method section. CASS18+ is a national network for professionals specialized in healthcare for autistic adults.

26. Provide a table with the demographic characteristics of the group.

We have provided all demographic information that we collected in lines 182 – 189 since we otherwise would have exceeded the Table limit of PLOS ONE. If the current editor is willing to make an exception in this case, we are happy to put the now typed out information into a table. 

27. Provide to the reader a brief explanation of Deserno et al., 2017 and also Frewen et al., 2012 study. Despite hypertextual material is interesting, this sort of tool is not suitable for PLOS ONE and burdens the reader.

We give a short description of the Deserno et al., 2017 study lines 80-84 and have now added a visual step-by-step explanation of the study in the Figure added in the context of reviewer comment below (page 10 of this document). We do not deem it relevant to the current manuscript to explain the study Frewen et al. (2012) conducted regarding anxiety, PTSD and depression. We do mention the study in the context of the presented methodology in lines 146-149: “Recent studies implementing this scaling technique have used it to get a self-reported representation of symptom-to-symptom interactions administered to individuals experiencing symptoms related to posttraumatic stress and anxiety (Frewen et al., 2012; 2013), repetitive behaviors (Ruzzano et al., 2015) and posttraumatic stress and eating disorders (Thornley et al., 2016).“ 

28. LINES 204 to 210: That is "Statistical Analysis" rather than "Measure and Procedure".

We thank the reviewer for spotting this and have moved this information to the Statistical Analysis part. 

29. FIGURE 1 and 2: The quality of the images does not allow us to read the texts inside the nodes or in the figure legends. Figure 2 needs a legend.

We regret to learn that the image quality was not sufficient. We will upload another format and hope that this will resolve the issue. Also, we have now added a legend to Figure 2.

30. LINES 243 – 247: The authors mention that: "Therefore, we decided to split the EMPIRICAL network in three (overlapping) parts with an (almost) equal number of nodes (j1= 14, j2=13, j3=13) based on their clustering in the EMPIRICAL network to ensure study feasibility."

Is that correct?

Yes, the reviewer is correct. We have changed the wording from empirical to ‘association network‘ now. 

31. LINES 243 – 252: This was the less comprehensible part of the methodology. Reformulate it to allow a better understanding. It would help a lot providing a diagram/draw/scheme with the step-by-step of the methodology.

We thank the reviewer for this suggestion and have created a scheme of our methodology. We hope this helps to clarify what the reported results are based on:

32. LINES 250 – 252: "Large rater groups"? The author assessed only 29 raters… It is not clear what was done. The low number of raters is a serious question. And, when allocated to one of the three different groups, the number is even smaller (n=9, 10, and 10).

Please see our response to the reviewer‘s previous remark no. 23. We apologize for causing confusion: For a rater study, based on the literature of structured interviews, the current study‘s sample size can be considered large (de Villiers et al., 2005) while we, of course, are aware that for empirical studies based on survey data this is considered small. We have now clarified this with the addition of lines 570 -573. 

33. LINE 259 – Why the authors have asked the raters to choose THREE targets? Based on what rationale?

This choice was based on the following rationale: We were aware that all factors included in the network somehow matter for well-being of autistic individuals, but we wanted the clinicians to make a selection of factors that they deem the most important intervention targets in the given network. In order to avoid the impression that we expected clinicians to rank-order the factors in the network, we limited the space of potential answers to three. The exact number of three answers was arbitrary but the goal was to limit the answers to avoid complete orderings of all factors. We have added this information to the manuscript in lines 260 - 261. Also, we planned to distribute the follow-up question regarding intervention targets in a second assessment round, i.e., after the rating assessments, as we wanted the intervention rating to be accompanied by a visual representation of the averaged clinician network. The intervention rating consisted of one simple question asking the participant to choose three intervention targets from the complete list of nodes depicted in the averaged clinician network. 

34. LINE 276 – Why the authors have used a threshold of 6? Based on what rationale?

Please see our response to the related comment below, comment no. 37.

35. It also is not clear why, having in mind that the empirical network reflects partial correlations scaled between -1 and 1, why the authors have used clinicians' ratings from 0 to 10. Why not using a Likert scale, for example? 

We thank the reviewer for the opportunity to clarify. Since the association network based on self-report data represents partial correlations, we decided on a continuous rating scale. An average of 7 on the rating scale would then be comparable to a partial correlation of 0.7. 

36. Moreover, comparing two different networks with two different methodologies would not be problematic?

We regret to say that we are not sure what the reviewer is asking here. The two networks we compare are based on another assessment method, and therefore differ in some regards. We discuss this difference in the methods section (e.g., line 301-306) and have presented three different approaches to comparing these network structures, each focusing on different aspects of the networks (i.e. absence/presence of edges, weights, integration as a scaffolding structure in network estimation). In these steps, our methodology is completely transparent in what we are comparing.

37. I wonder if the authors could have further commentaries on the following sentence from LINES 276 and 277:

—"Manually thresholding the visual representation was necessary since the raters tended to attribute very high values to edges, they thought were present."

Do the authors did not expect that ceiling effect (since they search for "perceived causal effects", those how seem to the raters more intense would be valued)? Do the way the questions were asked did not favor that behavior?

Moreover, in LINE 278, the authors mention that "did not specify such thresholds for any of the analyses". Nevertheless, they mention that (LINE 275-275) "only those causal relations endorsed by the raters with an average rating of at least 6 (on a scale from 0 to 10) on the PCR were included in the visual representation of the network". It seems that one sentence contradicts the other.

We thank the reviewer for pointing this out and have rephrased these sentences to avoid this confusion. We do specify a threshold in the visual representation of the network, i.e. the figure in the manuscript. Figure 2 depicts a regularized partial correlation network, which means that we used lasso regularization to control for Type 1 error, pushing small edges to zero. Since there is no similar technique for the rated network (as the connections are rated rather than estimated), we had to manually adjust for the fact that clinicians tended to attribute high values to edges they judged present (but not strong), resulting in a small, but high range, of edge weights. We did not, however, apply this threshold for further analyses. In the presented results all present edges are included, even if their average rating was smaller than 5. Furthermore, we do think that it was to be expected that the clinicians would rarely rate edges between these factors to be completely absent (i.e. 0). This is also in line with the theoretical assumption this whole endeavor is resting on: that these factors form a multicausal network of interrelations (see also our response to the last reviewer comment). 

### RESULTS

38. The results section is difficult to follow and, sometimes, no distinguished from what should be in the discussion. The sentences are very long, and the excessive number of examples and comparisons with previous studies (that should be done in Discussion rather than in Results — E.g. LINE 339 – 343) makes it difficult for the reader to retain the main findings among the profusion of information.

The text would benefit a lot if the authors would be more direct in reporting the results, commenting them only at the Discussion. Focus on the main findings. Although this sort of methodology provides uncountable information, is detrimental to the text if the authors try to mention everything they find.

We thank the reviewer for this suggestion and have reworked the results section with a focus on the main findings, moving the broader contextual information to the discussion section. 

39. The authors do not present numerical results in this section. This data is important and not presented only as supplementary material (LINE 359).

We thank the reviewer for this suggestion and have moved this information from the supplement to the main text, see 372-375.

40. All tables and figures must be close to their title, footnotes, and all material referent to them. Is difficult to follow the results because the figures do not present this information. Moreover, the Figures do not have titles, but long explanations of what they represent. That fact indicates clearly that the figures are not self-explanatory as they supposed to be (at least in partly).The quality of the images is poor making it impossible to read them.

We regret to learn that the figure conversion tool of editorial manager has decreased the quality of our figures which had a high resolution. Now we have included even higher-resolution image for all figures in our resubmission. We hope this resolves the issue. We have revisited all Figure captions to improve their clarity: 

41. LINE 337 – The authors should provide the list of abbreviations in the text and not only in the supplementary material.

We indeed solely provided an extensive list of the node abbreviations as supplementary material, but we agree with the reviewer that it is more convenient to have a Table next to the Figures listing all abbreviations used in the network and have, therefore, added such a list to the Figures directly (please see below).

42. The comprehension of the Tables and Figures should be independent of the text (and vice-versa). They have to contain all the necessary information for their comprehension (e.g. LINE 374 — the table title contains references to the text).

We thank the reviewer for this reminder and have adjusted all Figures and Tables to comply with the reviewer’s suggestion. 

43. There is no reference in the text regarding where Table 1 should be placed. It was put in the text but not mentioned.

We thank the reviewer for spotting this, and have added a reference for Table 1, considering the reviewer’s earlier comment. 

44. LINE 377 – 384: It is not clear why the authors decided to investigate the interventions the raters would do. That's not one of the aims of the study and with such a small number of raters, the results are not meaningful.

We understand the reviewer’s concern and hope that all the context provided above takes away the reviewer’s doubts regarding the sample size of the raters. We have now also added an explicit comment about this to the discussion section, see lines 582-585 and lines 610-612. 

45. LINE 387 – 393 and 458 – 469: This paragraph is not part of the Results section. It describes the methodology.

We appreciate the reviewer’s note. We added these short summaries of the methodological steps at the beginning of each of the three sub-reports so the reader knows what the following results are based on. We have now deleted the reminder in lines 387-393 and have moved the information added in lines 458-469 to the Methods section, lines 320-329. 

46. LINE 431: "…between -1 AND 1…"

We thank the reviewer and have corrected this typo.

47. Where is Figure 4 that is mentioned in the text?

We regret to learn that Figure 4 was not included in the reviewer’s manuscript version, and have thoroughly checked again that it is included in this submission. 

48. LINE 496 – 499: This paragraph is not part of the Results section and should be placed in the Discussion section.

We appreciate the reviewer’s suggestion and have removed this paragraph from the Results section.

### DISCUSSION

49. In general, the Discussion is easier to read than the rest of the text, although repetitive. Nevertheless, the authors tend to stretch the line in some conclusions to explain some findings. For example, I did not understand the sentence in LINE 526-528: "This would be consistent with the plausible idea that ASD symptoms arise from sources external to the current networks (e.g., from problems associated with brain development)."

How's that, "external to the current network"? Are not all the ASD symptoms part of a neurodevelopmental problem?

Moreover, having assessed only about 30 clinicians from a limited number of facilities is too forced, and — why not say — presumptuous, mention that "we found that the way clinicians perceive cause-effect relations between ASD symptoms".

We thank the reviewer for the opportunity to clarify. By ‘current networks’ we mean that there are more factors relevant to ASD than the ones included in the networks of this study. For example, early brain parameters are not assessed in the current study, but without any doubt relevant to atypical development (see Johnson et al., 2017 for an elaborate review of this line of thought). We have now explicitly added this information to line 545-549 in order to clarify what we mean by ‘external to the current network’. The reviewer’s second comment is in line with previous comments we have responded to in general comment no. 2. We have rephrased the sentence in line 563-565: “Third, we found that the way this clinicians sample perceive cause-effect relations between ASD symptoms, well-being, and domains of daily functioning is fairly similar to the interrelatedness of these factors found in self-reported data.“

50. Again, in Discussion, the problem of "causal" inferences persists.

Please see our response to the reviewer comment no. 2 above. 

51. I suggest the following articles in order to improve the quality of the text:

A. Docherty M and Smith R. The case for structuring the discussion of scientific papers. BMJ 1999;318;1224-1225;

B. Horton R. The hidden research paper. ‭JAMA,‭ June‭ ‬5,‭ ‬2002‭—‬Vol‭ ‬287,‭ ‬No.‭ ‬21‬‬.‬‬‬‬‬‬‬‬‬‬‬‬‬‬‬‬‬‬‬‬‬‬‬‬‬‬‬‬‬‬‬‬‬‬‬‬‬‬‬‬‬‬‬‬‬‬‬‬‬‬‬‬‬‬‬‬‬‬‬‬‬‬‬‬‬‬‬‬‬‬‬‬‬‬‬‬‬‬‬‬‬‬‬‬‬‬‬‬‬‬‬‬‬‬‬‬

We thank the reviewer for suggesting these articles and have re-structured the discussion section of our manuscript according to the suggested structure as much as possible. We have moved the section discussing weaknesses of the study up so that the discussion of its relation to the broader literature comes just before the section discussing unanswered questions and future research. 

52. LINE 523: Wouldn't be "…other than THE influence…" rather than "…other than THAT influence…"?‬‬‬‬‬‬‬‬‬‬‬‬‬‬‬‬

We thank the reviewer for spotting this and have changed our wording accordingly.

53. LINE 562: I do not think is fair the authors mention that they have used "large rater groups". The sample size is quite small and its subdivision smaller. This should be mentioned as a limitation.

Please see our response to similar comments above. 

54. LINE 568: It is interesting the authors mention that "it is important to note that the centrality of nodes in the empirical network indicates the importance of a node…", but they relegate centrality measures to supplementary material.

See our response to an earlier comment by the reviewer, we have now moved this information to the main text. 

55. LINE 572 – 573: This is one of the most important sentences in the manuscript that deserves more attention from the authors:

— "…assuming that the variables in question indeed form a causal system with symmetric effects…".

It is important to mention that, based on an assumption that variables investigated to form a causal system according to the current literature, the networks were built. So, it is necessary to caution regarding the conclusions of the study.

We would be keen to discuss this assumption more extensively with the reviewer and have now highlighted this again in the results section. Overall, we hope that our introduction of the network approach to psychology is a sufficient theoretical background for the reader to understand that this is the assumption under investigation. 

56. LINE 608: "schematic representation of PERCEPTION OF cause-effect " rather than "schematic representation of cause-effect "‬‬‬‬‬‬‬‬‬‬‬‬‬‬‬‬‬‬‬‬‬‬‬‬

We have changed our wording accordingly: 'The schematic representation of perceived cause-effect models might benefit both clinician and client in any mental health setting (see also Kroeze, 2013).'

---

## [Decision Letter · Decision Letter 1]

19 Nov 2020

Highways to happiness for autistic adults? Perceived causal relations among clinicians

PONE-D-20-15702R1

Dear Dr. Deserno,

We’re pleased to inform you that your manuscript has been judged scientifically suitable for publication and will be formally accepted for publication once it meets all outstanding technical requirements.

Kind regards,

Geilson Lima Santana, M.D., Ph.D.

Academic Editor

PLOS ONE

Additional Editor Comments (optional):

Reviewers' comments:

Reviewer's Responses to Questions

**Comments to the Author**

1. If the authors have adequately addressed your comments raised in a previous round of review and you feel that this manuscript is now acceptable for publication, you may indicate that here to bypass the “Comments to the Author” section, enter your conflict of interest statement in the “Confidential to Editor” section, and submit your "Accept" recommendation.

Reviewer #1: All comments have been addressed

2. Is the manuscript technically sound, and do the data support the conclusions?

Reviewer #1: Yes

3. Has the statistical analysis been performed appropriately and rigorously? 

Reviewer #1: Yes

4. Have the authors made all data underlying the findings in their manuscript fully available?

Reviewer #1: No

5. Is the manuscript presented in an intelligible fashion and written in standard English?

Reviewer #1: Yes

6. Review Comments to the Author

Reviewer #1: (No Response)

7. PLOS authors have the option to publish the peer review history of their article (what does this mean?). If published, this will include your full peer review and any attached files.

Reviewer #1: No

---

## [Editor Report · Acceptance letter]

26 Nov 2020

PONE-D-20-15702R1 

Highways to happiness for autistic adults? Perceived causal relations among clinicians 

Dear Dr. Deserno:

I'm pleased to inform you that your manuscript has been deemed suitable for publication in PLOS ONE. Congratulations! Your manuscript is now with our production department. 

Kind regards, 

on behalf of

Dr. Geilson Lima Santana 

Academic Editor

PLOS ONE